# KNOCKOUT: A SIMPLE WAY TO HANDLE MISSING INPUTS

## ABSTRACT

Deep learning models can extract predictive and actionable information from complex inputs. The richer the inputs, the better these models usually perform. However, models that leverage rich inputs (e.g., multi-modality) can be challenging to deploy widely, because some inputs may be missing at inference. Current popular solutions to this issue include marginalization, imputation, and training multiple models. Marginalization can obtain calibrated predictions, but it is computationally expensive and therefore only feasible for low dimensional inputs. Imputation may result in inaccurate predictions because it employs point estimates for missing variables and does not work well for high dimensional inputs (e.g., images). Training multiple models, where each model is designed to handle different subsets of inputs, can work well but requires knowing prior knowledge of missing input patterns. Furthermore, training and retaining multiple models can be costly. We propose an efficient method to learn both the conditional distribution using full inputs and the marginal distributions. Our method, Knockout, randomly replaces input features with appropriate placeholder values during training. We provide a theoretical justification for Knockout and show that it can be interpreted as an implicit marginalization strategy. We evaluate Knockout across a wide range of simulations and real-world datasets and show that it offers strong empirical performance.

## 1 INTRODUCTION

In many real-world applications of machine learning and statistics, not all variables might be available for every data point. This issue, as known as missingness, is well-studied in the literature (Little & Rubin, 2019) and common in fields like healthcare, social sciences, and environmental studies. From a Bayesian perspective, missingness can be viewed as a marginalization problem, where we would like a model to marginalize out the missing variables from the conditioning set. However, during training, we often do not know which features will be missing at inference time.

In lieu of training multiple models for every missingness pattern, a common strategy is imputation, which uses a point estimate (usually the mean or mode or a constant) to impute the missing feature (Le Morvan et al.). This can be seen as approximating the marginalization with a delta function. More sophisticated methods for handling missingness include using EM imputation (Josse et al., 2019) or neural-based imputation (Mattei & Frellsen, 2019; Ipsen et al., 2022). Although many prior methods may work well in some instances, they may not scale readily to high-dimensional inputs like images (Kyono et al., 2021; You et al., 2020), require additional networks for generation of missing variables (Ipsen et al., 2022), only apply to continuous inputs (Le Morvan et al., 2020; 2021), assume linearity of predictors (Le Morvan et al.), or make assumptions about the data distribution (Hazan et al., 2015).

In this work, we propose a simple, effective, and theoretically-justified augmentation strategy, called Knockout, for handling missing inputs. During training, features are augmented by randomly "knocking out" and replacing them with constant "placeholder" values. At inference time, using the placeholder value corresponds mathematically to estimation with the appropriate marginal distribution. In particular, we demonstrate how Knockout can be seen as implicitly maximizing the likelihood of a weighted sum of the conditional estimators and all desired marginals *in a single model*.

In a suite of experiments, we demonstrate the broad applicability of Knockout. We use both synthetic and real-world data with image-based and tabular inputs. Real world experiments include Alzheimer's forecasting, noisy label learning, multi-modal MR image segmentation and detection, and multi-view tree genus classification. We show the effectiveness of Knockout in handling low and high-dimensional missing inputs, and compare it with appropriate baselines, including imputation and ensemble-based methods.

## 2 METHOD

### 2.1 BACKGROUND

The goal of supervised machine learning (ML) is to learn the conditional distribution $p(Y|\boldsymbol{X})$ where $Y$ is the output (predictive target) and $\boldsymbol{X} \in \mathbb{R}^N$ are the vector of inputs or features. The prediction for a new sample $\boldsymbol{x}$ is $\hat{y} = \arg\max_Y p(Y|\boldsymbol{X} = \boldsymbol{x})$. However, in many practical applications, not all features may be present for a given input. Consider the case when $X_i$ is missing, and denote the vector of non-missing features as $\boldsymbol{X}_{-i}$. In general, multiple features may be missing at a time, and we can represent this with a missingness indicator set $\mathcal{M}$ and corresponding non-missing features as $\boldsymbol{X}_{-\mathcal{M}}$. In this case, what we really want is $p(Y|\boldsymbol{X}_{-\mathcal{M}})$.

How can we account for missingness? A simple approach is to train a separate model for $p(Y|\boldsymbol{X}_{-\mathcal{M}})$, i.e. a model that takes only the non-missing features $\boldsymbol{X}_{-\mathcal{M}}$ as inputs. However, this is expensive because a separate model is needed for each missingness pattern. Furthermore, there is no sharing of information between these separate models, even though they are theoretically related.

Another approach relies on rewriting $p(Y|\boldsymbol{X}_{-\mathcal{M}})$ in terms of the already available $p(Y|\boldsymbol{X})$:

$$p(Y|\boldsymbol{X}_{-\mathcal{M}}) = \int_{\boldsymbol{X}_{\mathcal{M}}} p(Y, \boldsymbol{X}_{\mathcal{M}}|\boldsymbol{X}_{-\mathcal{M}})d\boldsymbol{X}_{\mathcal{M}} = \int_{\boldsymbol{X}_{\mathcal{M}}} p(Y|\boldsymbol{X})p(\boldsymbol{X}_{\mathcal{M}}|\boldsymbol{X}_{-\mathcal{M}})d\boldsymbol{X}_{\mathcal{M}}. \quad (1)$$

The goal now is to obtain $p(\boldsymbol{X}_{\mathcal{M}}|\boldsymbol{X}_{-\mathcal{M}})$ and perform the integration over all possible $\boldsymbol{X}_{\mathcal{M}}$.

Imputation methods approximate Eq. 1 by replacing $p(\boldsymbol{X}_{\mathcal{M}}|\boldsymbol{X}_{-\mathcal{M}})$ with a delta function. For example, "mean imputation" uses the mean of the missing features $\boldsymbol{X}_{\mathcal{M}}$, $\mathbb{E}[\boldsymbol{X}_{\mathcal{M}}]$, for $\boldsymbol{X}_{\mathcal{M}}$ itself. In Eq. 1, this corresponds to approximating $p(\boldsymbol{X}_{\mathcal{M}}|\boldsymbol{X}_{-\mathcal{M}}) \approx \delta(\mathbb{E}[\boldsymbol{X}_{\mathcal{M}}])$, a delta function. While convenient and commonly used, mean imputation ignores the dependency between $\boldsymbol{X}_{\mathcal{M}}$ and $\boldsymbol{X}_{-\mathcal{M}}$, and does not account for any uncertainty.

More sophisticated approaches to imputation capture the interdependencies between inputs (Troyanskaya et al., 2001; Stekhoven & Bühlmann, 2012), for example by explicitly modeling $p(\boldsymbol{X}_{\mathcal{M}}|\boldsymbol{X}_{-\mathcal{M}})$ by training a separate model. At inference time, the point estimate $\boldsymbol{x}_{\mathcal{M}} = \arg\max_{\boldsymbol{X}_{\mathcal{M}}} p(\boldsymbol{X}_{\mathcal{M}}|\boldsymbol{X}_{-\mathcal{M}})$ can be used for the missing $\boldsymbol{X}_{\mathcal{M}}$. While properly accounting for interdependencies between inputs, this approach requires fitting a separate model for $p(\boldsymbol{X}_{\mathcal{M}}|\boldsymbol{X}_{-\mathcal{M}})$. In multiple imputation, multiple samples from $p(\boldsymbol{X}_{\mathcal{M}}|\boldsymbol{X}_{-\mathcal{M}})$ are drawn and a Monte Carlo approximation is used to estimate the integral on the RHS of Eq. 1 Kyono et al. (2021). Although this is more accurate than single imputation, it is not effective in high dimensional space.

### 2.2 KNOCKOUT

We propose a simple augmentation strategy for neural network training called Knockout that enables estimation of the conditional distribution $p(Y|\boldsymbol{X})$ and all desired marginals $p(Y|\boldsymbol{X}_{-\mathcal{M}})$ in a single, high capacity, nonlinear model, such as a deep neural network. During training, features are augmented by randomly "knocking out" and replacing them with constant, "placeholder" values. At inference time, using the placeholder value corresponds mathematically to estimation with the suitable marginal distribution.

Specifically, let $\boldsymbol{M} = [M_1, M_2, \ldots, M_N] \in \{0, 1\}^N$ denote a binary, induced missingness indicator vector. Let $\bar{\boldsymbol{x}} := [\bar{x}_1, \bar{x}_2, \ldots, \bar{x}_N] \in \mathbb{R}^N$ denote a vector of placeholder values. Then, define $\boldsymbol{X}'(\boldsymbol{M}, \boldsymbol{X}) = \boldsymbol{M} \odot \bar{\boldsymbol{x}} + (\boldsymbol{1} - \boldsymbol{M}) \odot \boldsymbol{X}$ as augmented Knockout inputs, where $\boldsymbol{1}$ is a vector of ones and $\odot$ denotes element-wise multiplication. During one training iteration, a different Knockout input is used corresponding to a different randomly sampled $\boldsymbol{M}$ for every data sample. The model weights are trained to minimize the loss function with respect to $Y$, as is done regularly.

Two mild conditions are required to ensure proper training. First, the placeholder values must be "appropriate," as we will elaborate below. For our theoretical treatment, we will use out-of support values as appropriate; i.e. $\bar{\boldsymbol{x}}_{\mathcal{M}} \notin \text{Support}(\boldsymbol{X}_{\mathcal{M}})$. Second, $\boldsymbol{M}$ must be independent of $\boldsymbol{X}$ and $Y$, i.e. $\boldsymbol{M} \perp\!\!\!\perp \boldsymbol{X}, Y$ [1] It follows straightforwardly that these two conditions lead to modeling the desired conditional and marginal distributions simultaneously. First, since $\bar{\boldsymbol{x}}_{\mathcal{M}}$ is not in the support of $\boldsymbol{X}_{\mathcal{M}}$,

$$\boldsymbol{X}'_{\mathcal{M}} = \bar{\boldsymbol{x}}_{\mathcal{M}} \iff \boldsymbol{M}_{\mathcal{M}} = \mathbf{1}, \qquad \boldsymbol{X}'_{\mathcal{M}} \neq \bar{\boldsymbol{x}}_{\mathcal{M}} \iff \boldsymbol{M}_{\mathcal{M}} = \mathbf{0} \text{ and } \boldsymbol{X}'_{\mathcal{M}} = \boldsymbol{X}_{\mathcal{M}}, \quad (2)$$

where $\mathbf{0}$ and $\mathbf{1}$ are vectors of zeros and ones of appropriate shape. Second, since $\boldsymbol{M}$ is independent of $\boldsymbol{X}$ and $Y$, it follows that imputing with the default value $\bar{\boldsymbol{x}}_{\mathcal{M}}$ is equivalent to marginalization of the missing variables defined by $\mathcal{M}$:

$$p(Y|\boldsymbol{X}'_{\mathcal{M}}=\bar{\boldsymbol{x}}_{\mathcal{M}}, \boldsymbol{X}'_{-\mathcal{M}}=\boldsymbol{x}_{-\mathcal{M}}) = p(Y|\boldsymbol{M}_{\mathcal{M}}=\mathbf{1}, \boldsymbol{M}_{-\mathcal{M}}=\mathbf{0}, \boldsymbol{X}_{-\mathcal{M}}=\boldsymbol{x}_{-\mathcal{M}}) = p(Y|\boldsymbol{X}_{-\mathcal{M}}=\boldsymbol{x}_{-\mathcal{M}}). \quad (3)$$

In particular, at the two extremes, no Knockout ($\boldsymbol{M} = \mathbf{0}$) corresponds to the original conditional distribution, and full Knockout ($\boldsymbol{M} = \mathbf{1}$) corresponds to the full marginal:

$$p(Y|\boldsymbol{X}'=\boldsymbol{x}) = p(Y|\boldsymbol{M}=\mathbf{0}, \boldsymbol{X}=\boldsymbol{x}) = p(Y|\boldsymbol{X}=\boldsymbol{x}), \quad (4)$$

$$p(Y|\boldsymbol{X}'=\bar{\boldsymbol{x}}) = p(Y|\boldsymbol{M}=\mathbf{1}) = p(Y) \quad (5)$$

For a new test input $\boldsymbol{x}$, the prediction when $\boldsymbol{x}_{\mathcal{M}}$ is missing is simply

$$\arg\max_Y p(Y|\boldsymbol{X}_{-\mathcal{M}}=\boldsymbol{x}_{-\mathcal{M}}) = \arg\max_Y p(Y|\boldsymbol{X}'_{\mathcal{M}}=\bar{\boldsymbol{x}}_{\mathcal{M}}, \boldsymbol{X}'_{-\mathcal{M}}=\boldsymbol{x}_{-\mathcal{M}}), \quad (6)$$

i.e., the learned estimator with the augmented Knockout input.

### 2.2.1 KNOCKOUT AS AN IMPLICIT MULTI-TASK OBJECTIVE

The missingness indicator $\boldsymbol{M}$ determines how inputs are replaced with appropriate placeholder values during training. To satisfy the independence condition of $\boldsymbol{M}$ with $\boldsymbol{X}$ and $\boldsymbol{Y}$, the variables $\boldsymbol{M}$ are sampled independently from a distribution $p(\boldsymbol{M})$ during training. We show that this training strategy can be viewed as a multi-task objective (Caruana, 1997) decomposed as a weighted sum of terms, where each term is a separate marginal weighted by the distribution of $\boldsymbol{M}$. Let $\ell$ denote the loss function to be minimized (e.g., mean-squared-error or cross-entropy loss):

$$L(\theta) = \mathbb{E}_{\boldsymbol{X}',Y}\, \ell(Y; f_\theta(\boldsymbol{X}'(\boldsymbol{M}, \boldsymbol{X}))) = \mathbb{E}_{\boldsymbol{X},Y}\mathbb{E}_{\boldsymbol{M}} \sum_{\boldsymbol{m} \in \boldsymbol{M}} \mathbb{I}(\boldsymbol{M}=\boldsymbol{m})\, \ell(Y; f_\theta(\boldsymbol{X}'(\boldsymbol{M}, \boldsymbol{X}))) \quad (7)$$

$$= \mathbb{E}_{\boldsymbol{X},Y} \sum_{\boldsymbol{m} \in \boldsymbol{M}} p(\boldsymbol{M}=\boldsymbol{m})\, \ell(Y; f_\theta(\boldsymbol{X}'(\boldsymbol{m}, \boldsymbol{X}))) \quad (8)$$

$$= \sum_{\boldsymbol{m}} p(\boldsymbol{M}=\boldsymbol{m})\, \mathbb{E}_{\boldsymbol{X},Y}\ell(Y; f_\theta(\boldsymbol{X}'(\boldsymbol{m}, \boldsymbol{X}))), \quad (9)$$

where $\mathbb{I}$ is the indicator function.

If there is knowledge about the missingness patterns at inference (e.g., some $X_i$ and $X_j$ exhibit correlated missingness), one can design $p(\boldsymbol{M})$ appropriately to cover all the expected missing patterns, i.e. by sampling $\boldsymbol{m}$ during training with different weights. In the absence of such knowledge, the most general distribution for $\boldsymbol{M}$ is i.i.d. Bernoulli. A common way correlated missingness arises in real-world applications is in structured inputs like latent features or images, where the entire feature vector or whole image is missing. In our experiments, we demonstrate the superiority of *structured* Knockout, over naive i.i.d. Knockout, when such correlated missingness is known a priori.

### 2.3 CHOOSING APPROPRIATE PLACEHOLDER VALUES

Our theoretical treatment assumes that the placeholder value $\bar{x}_i$ is not in the support of $X_i$ (see Appendix A.2 for further analysis). This is mathematically justified and works well in many cases, especially when $X_i$ is low dimensional. However, for high dimensional inputs like vectors/images, choosing an out-of-range placeholder can be suboptimal for practical reasons such as unstable gradients and/or limited modeling capacity. In the following sections, we relax the out-of-support assumption and make some recommendations for appropriate placeholder values for various types of $X_i$, informed by these practical considerations.

---

[1]Note it is not necessary that $M_i \perp\!\!\!\perp M_j$ for any $i, j$.

Table 1: List of different types of $X_i$ and the recommended $\bar{x}_i$

| Type of $X_i$ | Example | Dimension | Support | Normalized? | $\bar{\boldsymbol{x}}_i$ |
|---|---|---|---|---|---|
| Categorical | Gender | 1 | $\{1, \ldots, N_{X_i}\}$ | N/A | $N_{X_i} + 1$ |
| Continuous | Test scores | 1 | $[a, b]$ | Scale to [0, 1] | -1 |
| Continuous | Temperature | 1 | $[a, \infty)$ or $(-\infty, b]$ | Scale to $[0, \infty)$ | -1 |
| Continuous | White noise | 1 | $(-\infty, \infty)$ | Z-score | $\pm 10$ |
| Structured | Images | >1000 | $[a, b]$ | Scale to [0, 1] | **0** |
| Structured | Latent vectors | >16 | $(-\infty, \infty)$ | Z-score | **0** |

### 2.3.1 NON-STRUCTURED

In this section, we recommend suitable placeholder values for non-structured, scalar-valued inputs.

**Categorical.** If $X_i$ is a categorical variable with $N_{X_i}$ integer-valued classes from 1 to $N_{X_i}$, then $\bar{x}_i$ can be $N_{X_i} + 1$. If one-hot encoded, $\bar{x}_i$ can be a vector of 0s.

**Continuous and Non-empty Infeasible Set.** If $X_i$ is a continuous value within a bounded range, then we can scale the range to $[0, 1]$ and choose $\bar{x}_i = -1$. More generally, if $X_i$ has unbounded range but a non-empty infeasible set, then $\bar{x}_i$ can be set to a value in the infeasible set. For example, if $X_i$ only takes positive values, then we can set $\bar{x}_i = -1$.

**Continuous and Empty Infeasible Set.** When $X_i$ has unbounded range and an empty infeasible set, then we suggest applying $Z$-score normalization and choosing $\bar{x}_i$ such that it lies in a low probability region of the normalized $X_i$, $p(X_i = \bar{x}_i) \approx 0$. As we argue in Appendix A.1, this approach leads to an approximation of the desired marginal.

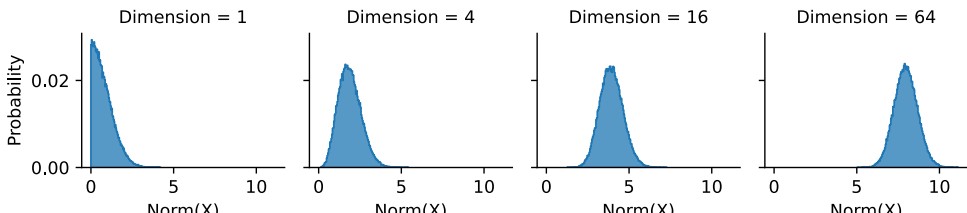

Figure 1: The region of high density of standard Gaussian shifts away from the origin as the number of dimensions increases. This motivates different choices of placeholder values at different dimensions.

Although, different distributions have different regions of low probability, we use the behavior of the Gaussian distribution as a guide to choose $\bar{x}_i$. Fig. 1 shows the histogram of the norm of points sampled from standard Gaussian distributions with different dimensionality. For a univariate standard Gaussian, most of the points lie close to the origin so we should choose $\bar{x}_i$ far away from the origin. However, as the dimension increases, most of the points lie on the hyper-sphere away from the origin so we should choose $\bar{x}_i$ to be the point at origin (i.e. a vector of zeros).

Table 1 summarizes the choices of $\bar{x}_i$ for different types of random variables $X_i$.

### 2.3.2 STRUCTURED

For structured inputs like images and feature vectors, we have found that Knockout applied with an out-of-support placeholder like $-1$, though theoretically sound, can cause issues like unstable gradients. Therefore, we recommend an appropriate placeholder to be either the image of all 0s or the mean image. When $Z$-score normalization is applied, the 0 image and the mean image coincide. Theoretically, it is well known that the mean of a high dimensional random variable, such as a Gaussian, has very low probability (Vershynin, 2018) (also see Fig. 1). We believe this

Table 2: Summary of experimental setups

| Task | Type of $X_i$ | Dimension | Normalized? | $\bar{x}_i$ | $\dot{x}$ |
|------|------|------|------|------|------|
| Simulations | Categorical/Continuous | 1 | Z-score (Cont. $X_i$) | 10 | -10 |
| Alzheimer's Forecasting | Continuous | 1 | Z-score | 10 | -10 |
| Privileged Information | Continuous | 1 | Scale to [0, 1] | -1 | N/A |
| Tumor Segmentation | Structured (images) | $256^3$ | Scale to [0, 1] | 0 | N/A |
| Tree Genus Classification | Structured (latent) | 768 or 2048 | Z-score | 0 | N/A |
| Prostate Cancer Detection | Structured (latent) | 256 | Z-score | 0 | N/A |

recommendation balances the tension between ensuring an extremely-low probability placeholder with proper convergence and performance. For an empirical demonstration, see Appendix A.3.

### 2.4 Observed Missingness during Training

The treatment above assumes complete training data, and inference-time missingness only. We now consider the situation where training data has *observed* missing inputs. Let $N$ be the binary mask indicating the observed data missingness. $N$ is different from $M$, which denotes the missingness induced by Knockout during training. Thus, $N$ is fixed for a data sample, while $M$ is stochastic. Observed missingness generally falls under the following scenarios (Little & Rubin, 2019).

**Missing Completely at Random (MCAR):** This implies that $N \perp\!\!\!\perp X, Y$. Let $M' := N \vee M$ be the augmented masking indicator, where $\vee$ denotes the logical OR operation. Since $N \perp\!\!\!\perp X, Y$ and $M \perp\!\!\!\perp X, Y$, so $M' \perp\!\!\!\perp X, Y$. Therefore, we can obtain the same result in Section 2.2 when using $M'$ instead of $M$ as the masking indicator vector. This implies that Knockout can be applied to MCAR training data simply by masking all the missing values using the same placeholders $\bar{x}$.

**Missing at Random (MAR) and Missing not at Random (MNAR):** This implies that $N \not\perp\!\!\!\perp X, Y$. Thus, we cannot replace the missing values in training data using the same placeholders. However, we can substitute these values using placeholders that are different from $\bar{x}$ but are also outside the support of the input variables (or very unlikely values). Let the placeholders for the data missingness be $\dot{x} \neq \bar{x}$. During training, Knockout still randomly masks out input variables, including those that are not observed in the data. Thus, the results in Section 2.2 still hold since $M \perp\!\!\!\perp X, Y$.

During inference, if we know a priori that $x_i$ of a sample is missing not at random, then we can use $\dot{x}_i$ as the placeholder. Otherwise, if we know $x_i$ is missing at completely random, we use $\bar{x}_i$.

## 3 Related Work

Knockout is similar to and inspired by other methods with unrelated aims. Dropout (Srivastava et al., 2014; Gal & Ghahramani, 2016) prevents overfitting by randomly dropping units (hidden and visible) during training and can be viewed as marginalizing over model parameters. During inference, marginalizing over parameters can be approximated by predicting once without dropout (Srivastava et al., 2014) or averaging multiple predictions with dropout (Gal & Ghahramani, 2016). Blank-out (Maaten et al., 2013) and mDAE (Chen et al., 2014) learn to marginalize out the effects of corruption over inputs. In contrast, Knockout learns different marginals to handle different missing input patterns.

Imputation techniques impute missing inputs explicitly, for example by imputing with the mean, median, or mode. In model-based imputation, a separate model or technique first predicts the missing inputs to impute. These models include k-nearest neighbors (Troyanskaya et al., 2001), chained equations (Van Buuren & Groothuis-Oudshoorn, 2011), random forests (Stekhoven & Bühlmann, 2012), autoencoders (Gondara & Wang, 2018; Ivanov et al., 2019; Lall & Robinson, 2022), GANs (Yoon et al., 2018; Li et al., 2019; Belghazi et al., 2019), or normalizing flows (Li et al., 2020). Although more accurate than simple mean/median imputation, model-based imputation incurs significant additional computation costs, especially when missing inputs are high-dimensional. In contrast, Knockout makes predictions without having to impute missing inputs explicitly. For example, some approaches (Ma et al., 2021; Peis et al., 2022) require additional training of multiple VAEs or sub-networks. Other approaches (Mattei & Frellsen, 2019; Ma et al., 2019) require training

only one VAE but they are formulated for homogeneous data (all continuous variables or all binary variables) and therefore not as flexible as Knockout. Besides, these approaches still require training two models (VAE and classifier), whereas Knockout trains only a single model (the classifier).

Another relevant line of work is causal discovery (Spirtes et al., 2000), which often involves fitting a model using different subsets of available inputs and multiple distributions simultaneously (Lippe et al., 2022; James et al., 2023). To reduce computational cost, it is common to train a single model that can handle different subsets of inputs using dropout (Ke et al., 2023; Brouillard et al., 2020; Lippe et al., 2022).

Techniques like Knockout are often used in practice to train a single neural network that models multiple distributions, but are often justified empirically with little care taken in choosing placeholder values. Many works use zeros without theoretical justification (Belghazi et al., 2019; Ke et al., 2023; Brouillard et al., 2020; Lippe et al., 2022). GAIN (Yoon et al., 2018) and MisGAN (Li et al., 2019) impute using out-of-support values similar to Knockout. However, both are limited in their treatment by assuming that the supports are bounded, and do not consider categorical variables. While the approach is similar to some prior work for structural inputs (Neverova et al., 2015; Parthasarathy & Sundaram, 2020) or low-dimensional inputs (Bertsimas et al., 2024), Knockout's theoretical backing shows that it can handle multiple data types and multiple missingness types (complete/MCAR/MAR/MNAR). Many self-supervised learning techniques can be interpreted as training to reconstruct the inputs with Knockout. In addition, Knockout can be trained with standard empirical risk minimization while some approaches need more complex optimization (Ma et al., 2021; 2022). For example, masked language modeling (Devlin et al., 2019) randomly maps tokens to an unseen "masked" token. Denoising autoencoders (Vincent et al., 2010) randomly replace image patches with black patches, which are arguably out of the support of natural images.

## 4 EXPERIMENTS

In all experiments, unless stated otherwise, we compare Knockout against a **common baseline** model trained on complete data, which, at inference time, imputes missing variables with mean (if continuous) or mode (if discrete) values. If the training is done on incomplete data with observed missing variables, imputed with mean/mode, we denote this as **common baseline\***. For most results we report a variant of Knockout but with sub-optimal placeholders (i.e. mean/mode for continuous/categorical features). We denote this variant as **Knockout\***. Note that both Knockout\* and common baseline\* use the same placeholder (mean/mode), with the only difference being that Knockout\*-trained models observe randomly knocked-out missingness *in addition to* (possible) observed missingness during training.

In all Knockout implementations, we choose random knockout rates such that, in expectation, half of the mini-batches have no induced missing variable. In batches with induced missingness, variables (or groups of variables in structured Knockout) are independently removed, with a probability equal to the knockout rate. The summary of the experimental setups are shown in Table 2.

### 4.1 SIMULATIONS

We perform simulations on both regression and classification, where the output $Y$ needs to be predicted from some input $X$. In each simulation run, we sample 30k data points in total and use 10% for training. All methods use the same neural network architecture composed of a 3-layer multi-layer perceptron (MLP) with hidden layers 100 and ReLU activations. Training is done using Adam (Kingma & Ba, 2014) with learning rate 3e-3 for 5k steps. We restrict our focus in this section to regression results. For further experimental details and classification experiments and results, see Appendix B.1. We generate training data corresponding to complete training data, MCAR training data, and MNAR training data. For MNAR data, we adopt the self-censored missing setup where a variable is missing if its value is above the variable 90th percentile. In the regression experiments, we additionally compare against missForest (Stekhoven & Bühlmann, 2012), a competitive baseline for inference-time imputation. We also include another popular baseline (ZI) which takes zero-imputed data and a missingness indicator/mask as inputs. We tried comparing against MIRACLE (Kyono et al., 2021) but the test set size (27k) and the high number of missing patterns tested make running MIRACLE intractable.

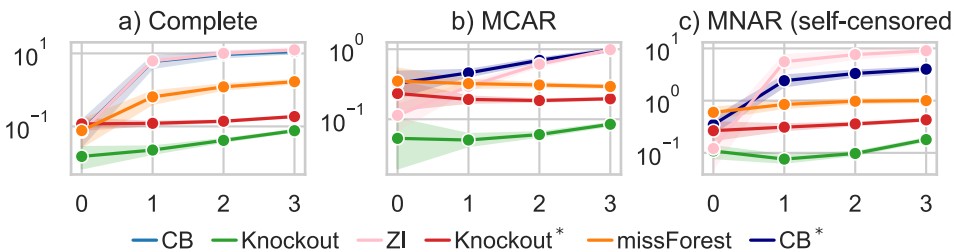

Figure 2: Test MSE evaluated against Bayes optimal prediction ($\mathbb{E}[Y|\boldsymbol{X}]$) from 10 repetitions of the regression simulation. Lower is better. X axis indicates the number of missing variables at inference time. CB: Common baseline, ZI: zero-imputation with mask. a) Complete training data. b) Missing completely at random (MCAR) training data. c) Missing not at random (MNAR) training data

We experiment on varying the number of missing features of $\boldsymbol{X} \in \mathbb{R}^9$ from 0 to 3. This resulted in 130 different missing patterns. We evaluate the models' predictions against the MMSE-minimizing Bayes optimal predictions: $\mathbb{E}[Y|\boldsymbol{X}]$. Fig. 2 shows the results of 10 repetitions of this simulation. Both variants of Knockout outperform baselines regardless of the types of training data (complete, MCAR, or MNAR). In particular, Knockout outperforms Knockout* in general; this underscores the importance of choosing an appropriate placeholder value.

### 4.2 MISSING CLINICAL VARIABLES IN ALZHEIMER'S DISEASE FORECASTING

We demonstrate Knockout's ability to manage observed missingness in a real-world clinical task: predicting the risk of progression from mild cognitive impairment (MCI) to Alzheimer's Disease (AD) over the next five years, using data from the Alzheimer's Disease Neuroimaging Initiative (ADNI) database (Mueller et al., 2005). Input features $\boldsymbol{X}$ include subject demographic variables, genetics, cognitive assessment scores, cerebrospinal fluid (CSF) measurements, and measurements derived from magnetic resonance imaging (MRI) and positron emission tomography (PET) images. The target $Y$ is a binary vector and indicates AD diagnosis in each of the five follow-up years. We employ the state-of-the-art model of Karaman et al. (2022). For Knockout, we use an out-of-range value of 10 for induced missingness and -10 for observed missingness, both during training and testing. Further details about the dataset and experimental setup are provided in Appendix B.2.

Figure 3 presents the average AUROC (area under the receiver operating characteristic curve) scores obtained when each input feature is missing during inference. We perform 10 random 80-20 train-test splits and calculate a Composite AUROC by averaging the AUROC scores from the five follow-up years in each split. We observe that Knockout outperforms the common baseline* in vast majority of cases, suggesting that knocking out input during training enhances the model's ability to handle missingness at test time. Furthermore, Knockout is largely better than Knockout*, which underscores the importance of choosing an appropriate placeholder. We note that we present a similar analysis using the complete portion of the training dataset (i.e., with no observed missingness in training data) in Figure S5 of Appendix B.2, further demonstrating Knockout's effectiveness.

### 4.3 PRIVILEGED INFORMATION FOR NOISY LABEL LEARNING

In this experiment, we show that Knockout can be used for learning with *privileged information* (PI) that is available in training but absent during testing. Specifically, we evaluate this in a noisy label learning task, where the objective is to use PI, such as annotator ID or annotation time, to enhance model robustness against label noise. Due to the absence of PI in testing, existing methods (Ortiz-Jimenez et al., 2023; Wang et al., 2023) require an auxiliary classification head for PI utilization. We demonstrate that Knockout can be directly applied with a method that accepts PI as input and achieve competitive performance. We follow previous experiment setups (Wang et al., 2023) and evaluated model performance on CIFAR-10H (Peterson et al., 2019) and CIFAR-10/100N (Wei et al., 2021). These datasets involve relabeled versions of the original CIFAR. For more details, see Appendix B.6.

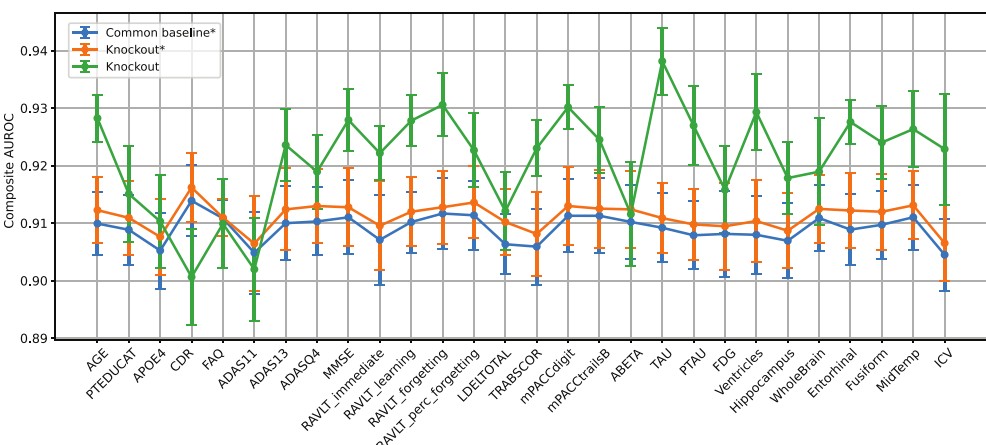

Figure 3: Composite AUROC scores obtained for the three model variants when each input feature is missing during inference (x-axis) in the Alzheimer's Disease forecasting experiment. Displayed are averages of 10 train-test splits. Error bars indicate the standard error across these splits.

Table 3: Test accuracy of different methods on noisy label datasets with PI. We report mean and standard deviation accuracy over 5 runs. PI quality "High" indicates a sample-wise PI is provided by the dataset. "Low" means only batch average is provided. Best results in **bold**, second-best underlined.

| Datasets | PI quality | No-PI | HET | SOP | Common baseline | Knockout |
|---|---|---|---|---|---|---|
| CIFAR-10H (Worst) | High | $51.1_{\pm 2.2}$ | $50.8_{\pm 1.4}$ | $51.3_{\pm 1.9}$ | $\underline{55.2_{\pm 0.8}}$ | $\mathbf{57.4_{\pm 0.6}}$ |
| CIFAR-10N (Worst) | Low | $80.6_{\pm 0.2}$ | $81.9_{\pm 0.4}$ | $\mathbf{85.0_{\pm 0.8}}$ | $82.3_{\pm 0.3}$ | $\underline{84.7_{\pm 0.7}}$ |
| CIFAR-100N (Fine) | Low | $60.4_{\pm 0.5}$ | $60.8_{\pm 0.4}$ | $\underline{61.9_{\pm 0.6}}$ | $60.7_{\pm 0.6}$ | $\mathbf{62.1_{\pm 0.3}}$ |

As a no-PI baseline, we train a Wide-ResNet-10-28 (Zagoruyko & Komodakis, 2016) model that ignores PI. We also compare against recent noisy label learning methods: HET (Collier et al., 2021) and SOP (Liu et al., 2022). We implement Knockout with a similar architecture and training scheme as the no-PI baseline, where we concatenate the PI with the image-derived features and randomly knock PI out during training. As a common baseline, we train the same architecture with complete training, but mean imputation for PI data during inference. Table 3 lists test accuracy results. For the CIFAR-10H dataset, where we have high quality PI, Knockout outperforms all baselines by a large margin, improving test accuracy by 6%. For CIFAR-10/100N datasets, where we have low quality PI during training, Knockout's boost is more modest, performing similarly with SOP and slightly better than HET and the no-PI baseline. We conclude that Knockout can offer competitive results when we have access to high quality PI during training.

### 4.4 MISSING IMAGES IN TUMOR SEGMENTATION

Here, we investigate the ability of Knockout to handle missingness in a high-dimensional, 3D dense image segmentation task. In particular, we experiment on a multi-modal tumor segmentation task (Baid et al., 2021), where the goal is to delineate adult brain gliomas in 3D brain MRI volumes given 4 modalities per subject: T1, T1Gd, T2, and FLAIR. We use a 3D UNet as the segmentation model (Ronneberger et al., 2015). We minimize a sum of cross-entropy loss and Dice loss with equal weighting and use Adam optimizer with a learning rate of 1e-3. See Appendix B.3 and A.3 for further details.

At inference time, we evaluate on all modality missingness patterns. Fig. 4 shows Dice scores. We observe that the Knockout-trained model has better Dice performance across all missingness

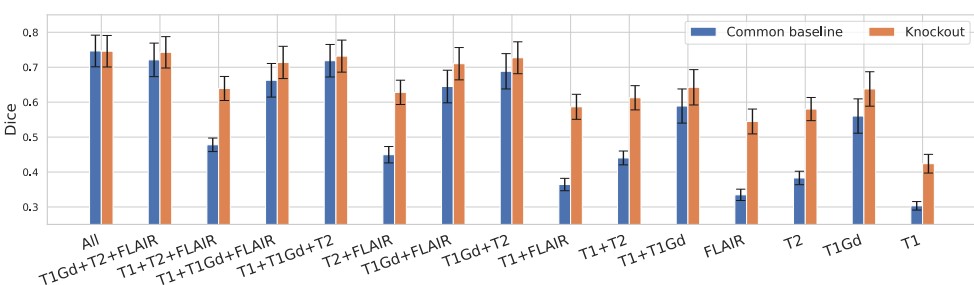

Figure 4: Dice performance of multi-modal tumor segmentation across varying missingness patterns of modality images. Knockout-trained models have better Dice performance across all missingness patterns than the common baseline. Error bars depict the 95% confidence interval over test subjects.

patterns. When all modalities are available, Knockout and the common baseline achieve the same performance level.

### 4.5 MISSING VIEWS IN TREE GENUS CLASSIFICATION

We demonstrate Knockout's ability to deal with missing data at the latent feature level in a classification task. The Auto Arborist dataset (Beery et al., 2022), a multi-view (street and aerial) image dataset, is used for this purpose. In this experiment, we used the top 10 genera for multi class prediction and reported results from 2 sites. A frozen ResNet-50 (He et al., 2016) and ViT-B-16 (Dosovitskiy et al., 2021) pretrained with ImageNet-v2 (Recht et al., 2019) is used as a feature extractor. The two features from street and aerial images are concatenated and were fed into 3-layer MLP with ReLU activations. We trained Knockout to randomly replace the whole latent vectors with vectors of 0s as placeholders after normalization. This variant is denoted as Knockout (Structured). We additionally trained two baselines for comparison: 1) Knockout (Features) where individual features in the latent vectors are independently replaced with placeholders, and 2) an imputation baseline, substituting latent vectors from missing views with vectors of zeros during inference. The results in Table 4 and Section S10 shows Knockout (Structured) outperforming Knockout (Features), suggesting that matching $p(M)$ with missing patterns that we expect to see at inference can be more effective.

Table 4: F1-scores of Auto Arborist averaged over 5 random seeds (site: Columbus). Each column represents non-missing modalities at inference time. Best results in **bold**, second-best underlined.

|  |  | Aerial+Street | Aerial | Street |
|---|---|---|---|---|
| ResNet-50 | Common baseline | $0.4834_{\pm 0.0167}$ | $\mathbf{0.3129}_{\pm 0.0177}$ | $0.3565_{\pm 0.0240}$ |
|  | Knockout (Features) | $\underline{0.4934}_{\pm 0.0209}$ | $0.2841_{\pm 0.0230}$ | $\underline{0.3814}_{\pm 0.0221}$ |
|  | Knockout (Structured) | $\mathbf{0.4961}_{\pm 0.0169}$ | $\underline{0.3089}_{\pm 0.0242}$ | $\mathbf{0.4165}_{\pm 0.0140}$ |
| ViT-B-16 | Common baseline | $0.4649_{\pm 0.0183}$ | $0.3052_{\pm 0.0224}$ | $\underline{0.3889}_{\pm 0.0110}$ |
|  | Knockout (Features) | $\underline{0.4732}_{\pm 0.0197}$ | $\underline{0.3159}_{\pm 0.0086}$ | $0.3833_{\pm 0.0108}$ |
|  | Knockout (Structured) | $\mathbf{0.4803}_{\pm 0.0179}$ | $\mathbf{0.3243}_{\pm 0.0196}$ | $\mathbf{0.4088}_{\pm 0.0151}$ |

### 4.6 MISSING MR MODALITIES IN PROSTATE CANCER DETECTION

We demonstrate structured Knockout in the context of a binary image classification task, where Knockout is applied at the latent level. The dataset consists of T2-weighted (T2w), diffusion-weighted (DWI) and apparent diffusion coefficient (ADC) MR images per subject (Saha et al., 2022). A simple "ensemble baseline" approach to address missingness is to train a separate convolutional classifier for each modality, and average the predictions of available modalities at inference time (Kim et al., 2023; Hu et al., 2020).

To train a model with latent-level structured Knockout, we use the same 3 feature extractors. Each feature extractor is trained with a different modality. The loss function is binary cross entropy loss and we use an Adam optimizer with a learning rate of 1e-3. We randomly knock out each modality. In the "common baseline" approach, we trained the same architecture with complete modalities. At inference time, the latent features from missing modalities are imputed with 0s. See Appendix B.5 for more details. In the "ensemble baseline" approach, we averaged the predicted values from the three extractors without additional training.

Knockout generally outperforms the baselines in the majority of scenarios as shown in Table 5 for F1 scores and Table S11 for AUC scores, except for inputs with ADC, where the common baseline achieves the best results. Notably, the F1 scores from the popular ensemble baseline are significantly lower than Knockout.

Table 5: F1 scores of prostate cancer dataset averaged over 5 random seeds, showing performance of ensemble baseline, common baseline, and Knockout across varying missingness patterns at inference time. Each column represents non-missing modalities. Best results in **bold**, second-best underlined.

| | T2 | ADC | DWI | ADC +DWI | T2 +DWI | T2 +ADC | All |
|---|---|---|---|---|---|---|---|
| Ensemble | $0.212_{\pm 0.091}$ | $0.373_{\pm 0.016}$ | $0.285_{\pm 0.032}$ | $0.327_{\pm 0.015}$ | $0.181_{\pm 0.044}$ | $0.337_{\pm 0.033}$ | $0.305_{\pm 0.050}$ |
| Common | $\underline{0.432}_{\pm 0.014}$ | $\mathbf{0.687}_{\pm 0.021}$ | $\underline{0.616}_{\pm 0.021}$ | $\mathbf{0.706}_{\pm 0.009}$ | $\underline{0.510}_{\pm 0.033}$ | $\mathbf{0.652}_{\pm 0.006}$ | $\underline{0.673}_{\pm 0.016}$ |
| Knockout | $\mathbf{0.639}_{\pm 0.023}$ | $\underline{0.601}_{\pm 0.019}$ | $\mathbf{0.628}_{\pm 0.025}$ | $\underline{0.677}_{\pm 0.016}$ | $\mathbf{0.667}_{\pm 0.010}$ | $\underline{0.649}_{\pm 0.023}$ | $\mathbf{0.688}_{\pm 0.014}$ |

## 5 CONCLUSION AND LIMITATIONS

We introduced Knockout, a novel, easy-to-implement strategy designed to handle missing inputs, using a mathematically principled approach. By simulating missingness during training via random "knock out" and substitution with appropriate placeholder values, our method allows a single model to learn the conditional distribution and all desired marginals. Our extensive experimental evaluation underscores the versatility and robustness of Knockout. Across diverse datasets, including both synthetic and real-world scenarios, Knockout consistently achieves competitive performance levels compared to conventional imputation and ensemble-based techniques across both low and high-dimensional missing inputs. We also extend Knockout to handle observed missing values in the training set. Our results highlight the importance of choosing the appropriate placeholder values for induced and observed missingness in training and during inference. Furthermore, we present structured version of Knockout that is more effective when entire feature vectors or input modalities might be missing.

There are several future directions for further investigation. While our paper highlights the importance of choosing an appropriate placeholder value, and there appears to be a practical tension between selecting an unlikely/infeasible value versus achieving numerical stability (e.g., avoiding exploding gradients), one can conduct a more detailed study of this to optimize the placeholder value. In our experiments, we did not compare Knockout with individual strong baseline models trained for specific missingness patterns. We considered this out of scope, as it became computationally infeasible for all the scenarios we considered. However, in practice, missingness patterns may be limited, making such an approach feasible. It remains unclear how Knockout would perform against such a strong baseline, which requires further evaluation. Another promising direction of future research is adapting Knockout to address distribution shifts in the presence of missingness. Finally, Knockout's theoretical treatment hinges on the use of a very high capacity, non-linear model trained on very large data. In applications, where low capacity models are used and/or training data are limited, Knockout might not be as effective.

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
