# A SUPPLEMENTARY MATERIAL

## A.1 PROOF OF CONTINUOUS AND UNBOUNDED CASE

As $p(X_i=\bar{x}_i) \approx 0$ implies $p(X_i=\bar{x}_i|\cdot) \approx 0$, as long as the conditioning is on a set of observations that is not extremely improbable. Then:

$$p(X_i'=\bar{x}_i|\cdot) = p(X_i'=\bar{x}_i, M_i = 1|\cdot) + p(X_i'=\bar{x}_i, M_i = 0|\cdot) \tag{10}$$

$$= p(M_i = 1)p(X_i'=\bar{x}_i|M_i = 1, \cdot) + p(M_i = 0)p(X_i'=\bar{x}_i|M_i = 0, \cdot) \tag{11}$$

$$= p(M_i = 1) + p(M_i = 0)p(X_i=\bar{x}_i|\cdot) \approx p(M_i = 1) \tag{12}$$

$$\Rightarrow p(Y|X_i'=\bar{x}_i, \cdot) = \frac{p(Y|\cdot)p(X_i'=\bar{x}_i|Y, \cdot)}{p(X_i'=\bar{x}_i|\cdot)} \approx \frac{p(Y|\cdot)p(M_i = 1)}{p(M_i = 1)} \approx p(Y|\cdot) \tag{13}$$

A similar analysis can be performed to derive the approximate versions of equations of Eq. (4) to (5), which are special cases.

## A.2 SUBOPTIMAL CHOICE OF PLACEHOLDER VALUES

If $\bar{x}_i \in \text{support}(X_i)$, then $X_i' = \bar{x}_i$ is either because of two mutually exclusive cases:

$$\begin{cases} M_i = 1 \text{ or} \\ M_i = 0 \text{ and } X_i = \bar{x}_i \end{cases}$$

Let $p(M_i = 0) = r$ and $\boldsymbol{x}_j \neq \bar{x}_j, \forall j \neq i$, then:

$$p(Y|X_i'=\bar{x}_i, \boldsymbol{X}_{-i}'=\boldsymbol{x}_{-i}) = p(Y|X_i'=\bar{x}_i, \boldsymbol{X}_{-i}=\boldsymbol{x}_{-i}) = \frac{p(Y, X_i'=\bar{x}_i|\boldsymbol{X}_{-i}=\boldsymbol{x}_{-i})}{p(X_i'=\bar{x}_i|\boldsymbol{X}_{-i}=\boldsymbol{x}_{-i})} \tag{14}$$

$$p(X_i'=\bar{x}_i|\boldsymbol{X}_{-i}=\boldsymbol{x}_{-i}) = p(M_i = 1) + p(M_i = 0, X_i=\bar{x}_i|\boldsymbol{X}_{-i}=\boldsymbol{x}_{-i}) \tag{15}$$

$$= 1 - r + p(X_i=\bar{x}_i|M_i = 0, \boldsymbol{X}_{-i}=\boldsymbol{x}_{-i})p(M_i = 0|\boldsymbol{X}_{-i}=\boldsymbol{x}_{-i}) \tag{16}$$

$$= 1 - r + r \times p(X_i=\bar{x}_i|\boldsymbol{X}_{-i}=\boldsymbol{x}_{-i}) \tag{17}$$

$$p(Y, X_i'=\bar{x}_i|\boldsymbol{X}_{-i}=\boldsymbol{x}_{-i}) = p(Y|\boldsymbol{X}_{-i}=\boldsymbol{x}_{-i})p(X_i'=\bar{x}_i|Y, \boldsymbol{X}_{-i}=\boldsymbol{x}_{-i}) \tag{18}$$

$$= p(Y|\boldsymbol{X}_{-i}=\boldsymbol{x}_{-i})(1 - r + r \times p(X_i=\bar{x}_i|Y, \boldsymbol{X}_{-i}=\boldsymbol{x}_{-i})) \tag{19}$$

From Eq. (17) and (19),

$$p(Y|X_i'=\bar{x}_i, \boldsymbol{X}_{-i}'=\boldsymbol{x}_{-i}) = p(Y|\boldsymbol{X}_{-i}=\boldsymbol{x}_{-i})\frac{1 - r + r \times p(X_i=\bar{x}_i|Y, \boldsymbol{X}_{-i}=\boldsymbol{x}_{-i})}{1 - r + r \times p(X_i=\bar{x}_i|\boldsymbol{X}_{-i}=\boldsymbol{x}_{-i})} \tag{20}$$

Thus, $p(Y|X_i'=\bar{x}_i, \boldsymbol{X}_{-i}'=\boldsymbol{x}_{-i}) = p(Y|\boldsymbol{X}_{-i}=\boldsymbol{x}_{-i})$ is equivalent to

$$\Leftrightarrow 1 - r + r \times p(X_i=\bar{x}_i|Y, \boldsymbol{X}_{-i}=\boldsymbol{x}_{-i}) = 1 - r + r \times p(X_i=\bar{x}_i|\boldsymbol{X}_{-i}=\boldsymbol{x}_{-i}) \tag{21}$$

$$\Leftrightarrow p(X_i=\bar{x}_i|Y, \boldsymbol{X}_{-i}=\boldsymbol{x}_{-i}) = p(X_i=\bar{x}_i|\boldsymbol{X}_{-i}=\boldsymbol{x}_{-i}) \tag{22}$$

$$\Leftrightarrow Y \perp\!\!\!\perp X_i=\bar{x}_i|\boldsymbol{X}_{-i}=\boldsymbol{x}_{-i} \tag{23}$$

It is difficult to find $\bar{x}_i$ to satisfy Eq. 23, most likely $p(Y|X_i'=\bar{x}_i, \boldsymbol{X}_{-i}'=\boldsymbol{x}_{-i}) \neq p(Y|\boldsymbol{X}_{-i}=\boldsymbol{x}_{-i})$.

## A.3 ANALYSIS OF PLACEHOLDERS FOR STRUCTURED KNOCKOUT

In this subsection, we analyze the effect of different placeholder values on a structured Knockout task, specifically the multi-modal tumor segmentation task from Section 4.4. We scale image intensity values to $[0, 1]$ per image. We train three Knockout models with the following placeholders: a constant image of -1s, a constant image of 0s, and the mean of all images per modality. At inference time, we evaluate on all modality missingness patterns. In the event that all images are missing, we randomly select one so that the model sees at least one image. For Knockout-trained models, the corresponding placeholder is imputed for missing images.

Fig. S1 shows the results. Interestingly, we observe the mean placeholder (Knockout*) performs better than constant-image placeholders, and the constant image of 0s generally outperforms the constant image of -1s. We hypothesize that in the context of structured inputs like images in conjunction with limited data and model capacity, placeholders which balance feasibility with practical considerations like causing unstable gradients due to out-of-range inputs is an important consideration.

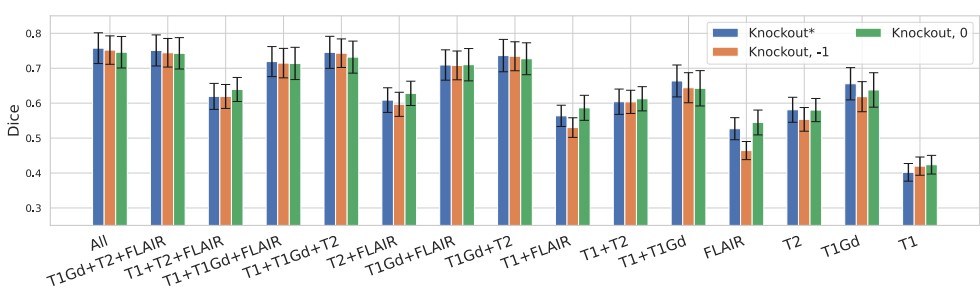

Figure S1: Dice performance of multi-modal tumor segmentation across varying missingness patterns of modality images. Knockout-trained models only. We observe mean placeholders perform better than constant-image placeholders. Error bars depict the 95% confidence interval over test subjects.

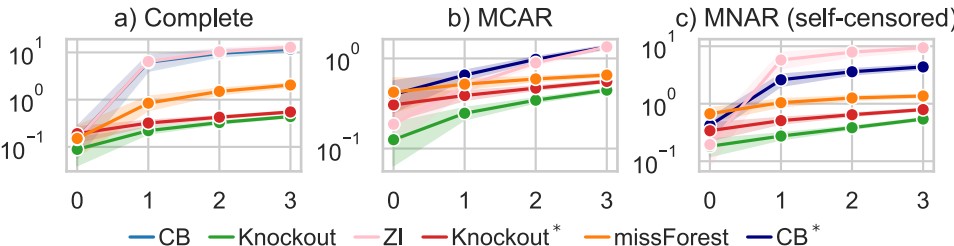

Figure S2: Test MSE evaluated against observations ($Y$) from 10 repetitions of the regression simulation. Lower is better. X axis indicates the number of missing variables at inference time. a) Complete training data. b) Missing completely at random (MCAR) training data. c) Missing not at random (MNAR) training data

## B EXPERIMENTAL DETAILS

All experiments were performed with access to a machine equipped with an AMD EPYC 7513 32-Core processor and an Nvidia A100 GPU. CPU testing was performed on a machine equipped with an Intel Xeon Gold 6330 CPU @ 2.00GHz. All code is written in PyTorch.

### B.1 SIMULATIONS

#### B.1.1 REGRESSION

In each repetition, the data are sampled from a 10-dimensional multivariate Gaussian distribution with mean $\boldsymbol{\mu}$ and covariance $\boldsymbol{\Sigma}$. The mean vector $\boldsymbol{\mu}$ is sampled uniformly from the interval $[0, 1]$, i.e. $\boldsymbol{\mu} \sim \text{Uniform}(0, 1) \in \mathbb{R}^{10}$. The covariance matrix is sampled as $\boldsymbol{\Sigma} := \boldsymbol{W}^T \boldsymbol{W}$, whereby $\boldsymbol{W} \sim \text{Uniform}(0, 1) \in \mathbb{R}^{10 \times 10}$. The first 9th variables of the multivariate Gaussian are assigned as $\boldsymbol{X}$ ($\boldsymbol{X} \in \mathbb{R}^9$) and the 10th variable is assigned as $Y$ ($Y \in \mathbb{R}$).

In addition to the MMSE-minimizing Bayes optimal predictions: $\mathbb{E}[Y|\boldsymbol{X}]$, we also evaluate the models' predictions against the observed values of $Y$ (Fig. S2). Since the input features have unbounded support, choosing appropriate placeholders (i.e. away from 0, see Fig. S3) is critical for getting good performance.

#### B.1.2 BINARY CLASSIFICATION

We evaluate the prediction error rate with full features ($\boldsymbol{X}$) and missing feature (only $X_1$ or $X_2$ as input). We also evaluate how close the models' predicted probability distributions with missing

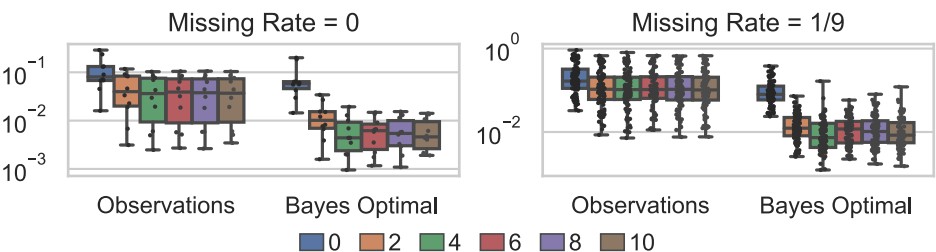

Figure S3: Test MSE in the regression simulation decreases as the placeholders move away from 0.

feature are against the marginal distributions (Fig. S4a and Fig. S4a top and left panels) using Jensen-Shannon divergence. The marginals distributions are estimated empirically using all data.

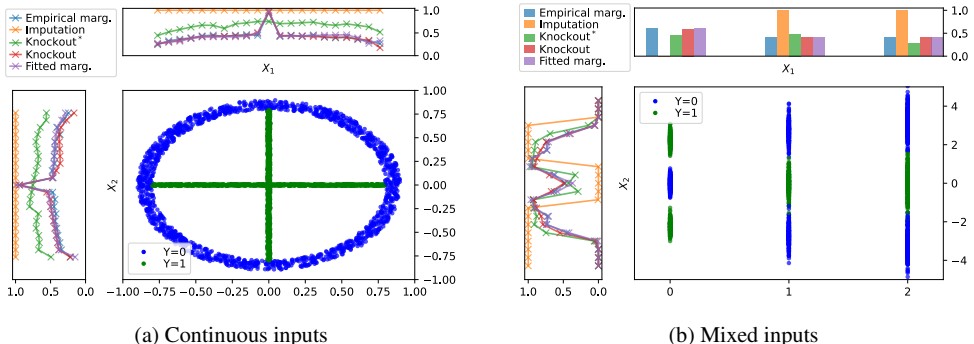

(a) Continuous inputs          (b) Mixed inputs

Figure S4: Visualization of the two classification simulations. Knockout's estimates of the marginal distributions (i.e. $P(Y|X_1)$ and $P(Y|X_2)$, denoted by red lines) are closer to the empirical estimates (blue lines) than baselines'. Top: $P(Y=1|X_1)$ estimated empirically and estimated by various approaches. Left: Various estimates of $P(Y=1|X_2)$. Bottom Right: Data visualization.

**Continuous Inputs.** All input variables are continuous ($X \in \mathbb{R}^2$). Knockout achieves similar error rate compared to standard training but much better performance when a variable is missing (Table S6). Knockout* performs worse than Knockout due to the sub-optimal choice of placeholders.

**Mixed Inputs.** $X$ consists of a binary variable and a continuous variable (i.e. $X_1 \in \{0, 1\}$, $X_2 \in \mathbb{R}$). Knockout achieves better results than baselines in all scenarios (Table S6).

## B.2 ALZHEIMER'S FORECASTING

All participants used in this work are from the Alzheimer's Disease Neuroimaging Initiative (ADNI) database. ADNI aims to evaluate the structure and function of the brain across different disease states and uses clinical measures and biomarkers to monitor disease progression. Applications for ADNI data use can be submitted through the ADNI website at https://adni.loni.usc.edu/data-samples/access-data/. Others would be able to access the data in the same manner as the authors. We did not have any special access privileges that others would not have. The investigators within the ADNI contributed to the design and implementation of ADNI and/or provided data but did not participate in analysis or writing of this report. Michael Weiner (E-mail: Michael.Weiner@ucsf.edu) serves as the principal investigator for ADNI.

We select the participants who have mild cognitive impairment (MCI) at the baseline (screening) visit and had at least one follow-up diagnosis within the next five years. We excluded participants who were diagnosed as CN in a later follow-up year (n=284) since these subjects might have been diagnosed incorrectly at some point. After this exclusion, we are left with 789 participants. Ta-

Table S6: Classification simulations. Best results are in bold. Err.: Proportion of test error. JSD: Jensen–Shannon divergence of the estimated and empirical marginal.

| Method | Missing Rate = 0 | Missing Rate = 1/2 | | | |
| --- | --- | --- | --- | --- | --- |
| | Err. ($X$) | Err. ($X_1$) | JSD ($X_1$) | Err. ($X_2$) | JSD ($X_2$) |
| Continuous inputs | | | | | |
| Common baseline | **0.0003** | 0.3970 | $\infty$ | 0.4001 | $\infty$ |
| Knockout* (Ours) | 0.0210 | 0.3549 | 0.0179 | 0.3727 | 0.0214 |
| Knockout (Ours) | 0.0007 | **0.2559** | **0.0003** | **0.2563** | **0.0007** |
| Fitted Marginals | N/A | 0.2531 | 0.0006 | 0.2600 | 0.0006 |
| Mixed inputs | | | | | |
| Common baseline | 0.0032 | 0.5972 | $\infty$ | 0.5410 | $\infty$ |
| Knockout* (Ours) | 0.0187 | 0.4843 | 0.0038 | 0.3410 | 0.0073 |
| Knockout (Ours) | **0.0031** | **0.4028** | **0.0001** | **0.2844** | **0.0009** |
| Fitted Marginals | N/A | 0.4028 | 0.0000 | 0.2809 | 0.0008 |

Table S7: Summary statistics of the participants at baseline in the Alzheimer's Disease data. Mean ± standard deviations are listed. APOE4 row represents the number of alleles.

| Characteristic | ($n$=789) |
| --- | --- |
| Female/Male | 324/465 |
| Age ($yr$) | $73.46 \pm 7.39$ |
| Education ($yr$) | $15.93 \pm 2.81$ |
| APOE4 (0/1/2) | 371/313/98 |
| CDR | $1.55 \pm 0.89$ |
| MMSE | $27.52 \pm 1.82$ |

ble S7 lists summary statistics for the participants; including sex, age, number of years of education completed, count of Apolipoprotein E4 (APOE4) allele, Clinical Dementia Rating(CDR), and Mini Mental State Examination (MMSE) scores at baseline.

As is common in many real-world longitudinal studies, ADNI experiences missing follow-up visits, irregular timings, and high dropout rates before the study's planned end. Table S8 shows the number of subjects available in each diagnostic category for annual follow-ups. In Table S8 and all analyses, any subject who progressed from MCI to AD before withdrawing was considered to remain in the AD state until the fifth year, reflecting AD's irreversible nature. We employed the reweighted cross-entropy loss scheme introduced in Karaman et al. (2022) during training to account for the imbalance in diagnoses.

Our input features include subject demographics (age; and number of years of education completed, or PTEDUCAT), genotype (number of APOE4 alleles), cognitive assessments (Clinical Dementia Rating, or CDR; Activities of Daily Living, or FAQ; Alzheimer's Disease Assessments 11, 13, and Q4, or ADAS11, ADAS13, ADASQ4, respectively; Mini-Mental State Exam, or MMSE; Rey Auditory Verbal Learning Test Trials, or RAVLT immediate, learning, forgetting, and percent forgetting;

Table S8: The number of available subjects in each diagnostic group for annual follow-up visits in the Alzheimer's Disease data. The follow-up diagnoses are not actually exactly 12 months apart. They have been rounded to the nearest time horizon in years.

| Follow-up year | 1 | 2 | 3 | 4 | 5 |
| --- | --- | --- | --- | --- | --- |
| MCI | 674 | 431 | 317 | 202 | 127 |
| AD | 110 | 218 | 261 | 286 | 292 |

Table S9: The degree of missingness (%) in different data modalities in the Alzheimer's Disease data.

| Data Type | Missingness Rate (%) |
|---|---|
| Demographics | 0.06 |
| Genotype | 0.89 |
| Cognitive assessments | 0.20 |
| CSF | 37.14 |
| MRI | 21.22 |
| FDG | 24.08 |

Logical Memory Delayed Recall, or LDELTOTAL; Trail Making Test Part B, Or TRABSCOR; and Digit and Trails B versions of Preclinical Alzheimer's Cognitive Composite score, or mPACCdigit and mPACCtrailsB, respectively). The biomarkers are Cerebrospinal Fluid (CSF) measurements (Amyloid-Beta 1–42, or ABETA; Total Tau, or TAU; Phosphorylated Tau, or PTAU), Magnetic Resonance Imaging (MRI) volume measurements (Ventricles; Hippocampus; WholeBrain; Entorhinal; Fusiform; MidTemp; Intracranial Volume, or ICV; all computed using the FreeSurfer software (Desikan et al., 2006; Fischl et al., 2004)), and Positron Emission Tomography (PET) standardized uptake value ratio (SUVR) score for tracer Fluorine-18-Fluorodeoxyglucose, or FDG. We note that all of our input features are numerical and we perform z-score normalization using the mean and standard deviation values derived from training data. The degree of missingness for each modality in the dataset can be seen in S9.

We note that we train our models using the hyperparameters stated in Karaman et al. (2022). Figure S5 shows the Composite AUROC scores obtained using the complete portion of the dataset ($n = 256$ subjects). In this experiment, the training data has no observed missing variables. These results are similar to the results included in the main text, where Knockout outperforms the baseline and the choice of the appropriate placeholder has an impact on the performance.

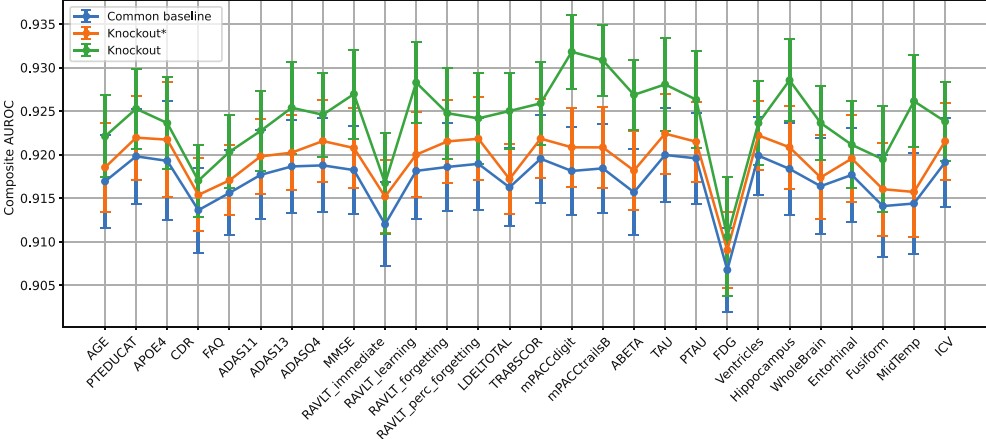

Figure S5: Composite AUROC scores obtained for the three model variants when each input feature is missing during inference (x-axis) for the complete data case in the Alzheimer's Disease forecasting experiment. Displayed are averages of 10 train-test splits. Error bars indicate the standard error across these splits.

## B.3 MULTI-MODAL TUMOR SEGMENTATION

The RSNA-ASNR-MICCAI BraTS Baid et al. (2021) challenge releases a dataset of 1251 subjects with multi-institutional routine clinically-acquired multi-parametric MRI scans of glioma. Each subject has 4 modalities: native (T1), post-contrast T1-weighted (T1Gd), T2-weighted (T2), and T2 Fluid Attenuated Inversion Recovery (T2-FLAIR). All the imaging datasets have been annotated

manually, by one to four raters, following the same annotation protocol, and their annotations were approved by experienced neuro-radiologists. Annotations comprise the GD-enhancing tumor (ET — label 3), the peritumoral edematous/invaded tissue (ED — label 2), and the necrotic tumor core (NCR — label 1).

The following pre-processing is applied: co-registration to the same anatomical template, interpolation to the same resolution (1 mm3), skull-stripped, and min-max normalized to the range [0, 1]. The ground truth data were created after their pre-processing. For training, we use 80%/5%/15% data split of the subjects for training/validation/testing.

For the segmentation model, we use a 3D UNet with 4 downsampling layers and 2 convolutional blocks per resolution (Ronneberger et al., 2015). We minimize a sum of cross-entropy loss and Dice loss with equal weighting and use an Adam optimizer with a learning rate of 1e-3.

## B.4 TREE GENUS CLASSIFICATION

Table S10 shows the F-1 scores of the second site in the Tree Genus Classification experiment.

Table S10: F1-scores of Auto Arborist averaged over 5 random seeds (site: Buffalo). Best results in **bold**, second-best underlined.

|  | Aerial+Street | Aerial | Street |
|---|---|---|---|
| Common baseline | $\mathbf{0.3684}_{\pm 0.0108}$ | $\underline{0.1871}_{\pm 0.0228}$ | $\underline{0.2969}_{\pm 0.0447}$ |
| Knockout (Features) | $0.3367_{\pm 0.0255}$ | $0.1702_{\pm 0.0204}$ | $0.2408_{\pm 0.0283}$ |
| Knockout (Structured) | $\underline{0.3585}_{\pm 0.0219}$ | $\mathbf{0.1997}_{\pm 0.0155}$ | $\mathbf{0.3539}_{\pm 0.0135}$ |

## B.5 PROSTATE CANCER DETECTION

A common clinical workflow for the diagnosis of prostate cancer is to detect and localize abnormalities from 3 MR modalities: T2-weighted (T2w), diffusion-weighted (DWI) and apparent diffusion coefficient (ADC) images (Turkbey et al., 2019). T2w images provide anatomical details, while DWI and ADC highlight restricted diffusion, which can be a sign of malignancy.

We divided 1500 biparametric MR image sets provided from Prostate Imaging: Cancer AI (PICAI) challenge (Saha et al., 2022) "training" dataset into training, validation, test sets in a 0.6/0.2/0.2 ratio. Among the 1500 cases, 425 were confirmed as cancer by biopsy. DWI and ADC images are registered to T2w images and all images are cropped around prostate and resized to $100 \times 100 \times 40$. For the modality-wise classification tasks, we used 3D CNN with 4 blocks, each with a convolution layer, BatchNorm, leakly ReLU activation and average pooling layer, followed by fully connected layer. We trained the models to predict PCa using binary cross entropy loss and an Adam optimizer with a learning rate of $1e - 3$.

Table S11: AUC performance for prostate cancer detection from the ensemble baseline, common baseline, and Knockout, across varying missingness patterns at inference time. Each column represents non-missing modalities. Best results in **bold**, second-best underlined.

|  | T2 | ADC | DWI | ADC +DWI | T2 +DWI | T2 +ADC | All |
|---|---|---|---|---|---|---|---|
| Ensemble | $0.683_{\pm 0.013}$ | $\mathbf{0.786}_{\pm 0.010}$ | $0.718_{\pm 0.007}$ | $0.780_{\pm 0.005}$ | $0.722_{\pm 0.005}$ | $\underline{0.766}_{\pm 0.006}$ | $\underline{0.766}_{\pm 0.004}$ |
| Common | $\underline{0.687}_{\pm 0.011}$ | $\underline{0.771}_{\pm 0.011}$ | $\underline{0.720}_{\pm 0.007}$ | $\underline{0.784}_{\pm 0.003}$ | $\underline{0.727}_{\pm 0.006}$ | $\mathbf{0.771}_{\pm 0.008}$ | $\mathbf{0.774}_{\pm 0.004}$ |
| Knockout | $\mathbf{0.694}_{\pm 0.009}$ | $0.730_{\pm 0.019}$ | $\mathbf{0.736}_{\pm 0.009}$ | $\mathbf{0.789}_{\pm 0.005}$ | $\mathbf{0.744}_{\pm 0.004}$ | $0.753_{\pm 0.011}$ | $\mathbf{0.774}_{\pm 0.007}$ |

## B.6 PRIVILEGED INFORMATION FOR NOISY LABEL LEARNING

We briefly introduce two datasets we used for this experiment: CIFAR-10H (Peterson et al., 2019) and CIFAR-10/100N (Wei et al., 2021). CIFAR-10H relabels the original CIFAR-10 10K test set

| Datasets | PI quality | SOP | Common baseline | Knockout* | Knockout |
|---|---|---|---|---|---|
| CIFAR-10H (Worst) | High | $51.3_{\pm 1.9}$ | $55.2_{\pm 0.8}$ | $56.9_{\pm 0.59}$ | $\mathbf{57.4_{\pm 0.6}}$ |
| CIFAR-10N (Worst) | Low | $\mathbf{85.0_{\pm 0.8}}$ | $82.3_{\pm 0.3}$ | $83.6_{\pm 0.7}$ | $84.7_{\pm 0.7}$ |
| CIFAR-100N (Fine) | Low | $61.9_{\pm 0.6}$ | $60.7_{\pm 0.6}$ | $61.6_{\pm 0.6}$ | $\mathbf{62.1_{\pm 0.3}}$ |

Table S12: Test accuracy of different methods on noisy label dataset with PI. Best results in **bold**, second-best underlined.

with multiple annotators and provides high-quality sample-wise annotation information such as annotator ID, reaction time and annotator confidence as PI. Following a previous setup (Wang et al., 2023), we test on the high-noise version of CIFAR-10H, by selecting incorrect labels when available, denoted as "CIFAR-10H Worst". The estimated noise rate is 64.6%. While we train on the high-noise version, testing is conducted on the original CIFAR-10 50K training set. CIFAR-10/100N provides multiple annotations for CIFAR-10/100 training set. The raw data also includes information about annotation process. But this information is provided as averages over batches of examples rather than sample-wise. The estimated noise rate is 40.2% for CIFAR-10/100N.

For all CIFAR experiments and baselines, we use the Wide-ResNet-10-28 (Zagoruyko & Komodakis, 2016) architecture. We use SGD optimizer with 0.9 Nesterov momentum, a batch size of 256, 0.1 learning rate and 1e-3 weight decay and minimized the cross-entropy loss with respect to the provided labels. The total training epoch is 90, and the learning rate decayed by a factor of 0.2 after 36, 72 epochs. For the PI features, we use annotator ID and annotation reaction time. In PI features are normalized to [0, 1] for preprocessing. For Knockout, during training, we randomly knock out all PI features at 50% rate and use -1 as placeholder value. All experiments are performed on one A6000. In Table S12, We further show results for common baseline, Knockout* and Knockout.