# OpenReview forum: "Knockout: A simple way to handle missing inputs"
_ICLR.cc/2025/Conference — ICLR 2025 Conference Withdrawn Submission_

### Official Review · Reviewer_L32p · 2024-10-23

**Soundness:** 3
**Presentation:** 3
**Contribution:** 2
**Rating:** 3
**Confidence:** 3

**Summary:**

This paper proposes a simple technique, Knockout, which randomly replaces input features with appropriate placeholder values during training. The main contribution/novelty of this paper is to introduce specialized placeholder values for different types of features: categorical, bounded vs unbounded continuous, structured vs unstructured.

**Strengths:**

I appreciate the authors efforts to run experiments on various datasets.
The paper is also interesting and practical, proposing a simple and straightforward technique.

**Weaknesses:**

Although this paper is interesting and practical. But, it is very incremental in terms of research novelty considering the expectations from an ICLR paper. For these types of papers, it is required to have thorough experimental studies and solid comparisons to show the applied contributions. But, I think this paper has lack of comparisons with important baselines or prior works.

* The main missing baseline for comparison is the dropout method. Actually, the comparison between knockout and knockout* shows that most of the advantage comes from random replacement. In fact, since the model has seen more missing data during training, it has become more robust to missingness and performs better on test data. An intuitive baseline is to apply dropout with the same missing rate and compare with knockout. Likewise, it may be possible that by applying the same rate of random missingness and use any other baseline (e.g. mean/media imputation or other methods), their performance also improves. In general, I think the comparisons performed in the paper were not fair since the other methods did not see the same amount of missing values during training

* The paper misses to cite some important related works:
    * Why not to use zero imputation? correcting sparsity bias in training neural networks
    * Debiasing Averaged Stochastic Gradient Descent
    * Learning from data with structured missingness

* Further, the paper cites some important papers such as "Ipsen et al, How to deal with missing data in supervised deep learning" but does not perform any comparison. I understand this paper proposes a complicated model. But, for an ICLR paper, I think stronger baseslines are expected.

**Questions:**

I explained my concerns in the weaknesses section.
I may change my rating after rebuttal and discussion with other reviewers.

---

### Official Review · Reviewer_wHDa · 2024-10-28

**Soundness:** 3
**Presentation:** 3
**Contribution:** 3
**Rating:** 6
**Confidence:** 3

**Summary:**

This paper introduces "Knockout," a simple yet theoretically-grounded method for handling missing input features in deep learning models. The key idea is to randomly replace input features with appropriate placeholder values during training, which enables a single model to learn both the conditional distribution using full inputs and the marginal distributions for cases with missing inputs. The authors provide theoretical justification showing that Knockout can be interpreted as an implicit marginalization strategy and demonstrate its effectiveness across diverse scenarios including synthetic data, clinical predictions, noisy label learning, tumor segmentation, and multi-modal classification tasks. Compared to existing approaches like marginalization (computationally expensive), imputation (potentially inaccurate for high-dimensional data), or training multiple models (costly and requires prior knowledge of missing patterns), Knockout offers a more efficient and flexible solution. The authors also carefully analyze how to choose appropriate placeholder values for different types of inputs and show how Knockout can handle both complete and incomplete training data under various missingness mechanisms (MCAR, MAR, MNAR).

**Strengths:**

- Elegant and practical solution that balances simplicity with theoretical soundness
- Strong theoretical foundation with rigorous mathematical analysis
- Impressive versatility across different data types and applications
- Practical single-model solution compared to existing multi-model approaches
- Comprehensive empirical evaluation with meaningful baselines
- Clear and actionable implementation guidelines for practitioners

**Weaknesses:**

- Limited theoretical analysis for finite-capacity models and small datasets, as theory assumes high-capacity models and large data
- Missing comparison against specialized models trained for specific missingness patterns
- No detailed ablation study on optimal placeholder value selection, despite its importance
- Lack of exploration into computational overhead during training compared to simpler approaches
- Limited discussion of failure cases or scenarios where the method might underperform
- No investigation into potential impact on model robustness or calibration

**Questions:**

- How does Knockout conceptually and empirically compare to related data augmentation methods like Dropout and DAMP [1], particularly since all these methods involve randomly masking inputs during training?
- How does the method perform with limited training data or when using simpler model architectures?
- What is the computational overhead during training compared to standard training and other baselines?
- How sensitive is the performance to the choice of placeholder values in practice?
- Could the random knockout process affect model calibration or uncertainty estimates?
- When would traditional imputation methods be preferable to this approach?
- Have the authors considered extending this to sequential data or other structured input types?

1. Trinh et al. Improving robustness to corruptions with multiplicative weight perturbations. NeurIPS 2024.

---

### Official Review · Reviewer_xNRN · 2024-11-02

**Soundness:** 3
**Presentation:** 4
**Contribution:** 3
**Rating:** 6
**Confidence:** 3

**Summary:**

The paper presents Knockout, a strategy for handling missing inputs in complex models. Knockout randomly replaces input features with placeholders during training, enabling a single model to learn conditional and marginal distributions. This method is theoretically sound and intensive simulation and real data application were used to evaluate the performance of Knockout.

**Strengths:**

1. The manuscript is generally well-written, demonstrating quality and clarity.
2. The author presents a comprehensive review of related work.
3. The new Knockout method is evaluated against multiple strong baselines.
4. The author thoroughly discusses various types of missing data mechanisms and evaluates the performance of Knockout and common baselines on them.

**Weaknesses:**

1. Figure 2 shows that selecting an appropriate placeholder value has a strong impact on Knockout. While the author emphasizes the importance of this choice, a general guideline for choosing placeholder values is lacking, leaving it to be determined on a case-by-case basis.
2. The simulation results appear somewhat limited. The input dimension of X is only 9, and the number of missing features ranges from 0 to 3. It would be beneficial to include simulations that better align with real-world dat. Specifically, those with high dimensionality and higher missing rates.

**Questions:**

I believe this paper is well-written and organized, so I don’t have any direct questions. However, I am curious if the authors have considered applying the Knockout method to every hidden layer of the neural network, rather than just the input layer. This method seems somewhat similar to dropout, which is typically applied across all layers.

---

### Official Review · Reviewer_sKPi · 2024-11-03

**Soundness:** 3
**Presentation:** 3
**Contribution:** 2
**Rating:** 3
**Confidence:** 4

**Summary:**

The paper introduces "Knockout," a simple yet effective data augmentation strategy for handling missing inputs during inference in machine learning models. Knockout randomly replaces input features with appropriate placeholder values during training. At inference time, using the placeholder value corresponds to marginalization over the missing variables.

The key contributions are:

1) Theoretical justification showing that Knockout implicitly maximizes the likelihood of a weighted sum of the conditional estimators and all desired marginals in a single model.

2) Analysis and recommendations for choosing appropriate placeholder values for different data types (categorical, continuous, structured).

3) Extensive experiments on synthetic and real-world datasets (images, tabular data) demonstrating Knockout's effectiveness in handling missing inputs across various scenarios, outperforming common baselines like mean/mode imputation and ensemble methods.

**Strengths:**

- The theoretical analysis is well-reasoned, and the empirical evaluation is comprehensive, covering a diverse set of tasks and data modalities. The authors have taken care to compare against appropriate baselines and provide ablation studies (e.g., structured vs. unstructured Knockout).
- The paper is well-written and clearly explains the core idea, theoretical justification, and experimental setup. The authors have provided sufficient details to facilitate reproducibility.

**Weaknesses:**

- The idea of randomly masking/corrupting inputs during training is not entirely new, many papers in related work section essentially use the same approach, eg PartialVAE, VAEAC, ACFlow,
- While the authors provide theoretical justification for Knockout, the analysis relies on the assumption of using a very high capacity, non-linear model trained on large data. It is unclear how well Knockout would perform in scenarios with limited data or low-capacity models.
- The comparison against strong baselines trained specifically for certain missingness patterns is missing. In practical scenarios where the missingness patterns are known or limited, such specialized models could potentially outperform Knockout.

**Questions:**

- The choice of the placeholder value seems crucial for Knockout's performance. While the authors provide recommendations, have you explored techniques to automatically learn or optimize the placeholder values during training?
- In scenarios with limited training data or low-capacity models, how well does Knockout perform compared to simpler baselines like mean/mode imputation? Are there any modifications or variations of Knockout that could improve its effectiveness in such cases?
- Have you explored the potential of Knockout to handle distribution shifts in the presence of missingness? For example, if the missingness patterns or distributions change between training and inference, how would Knockout perform compared to other methods?
- While the paper covers a diverse set of tasks and data modalities, it would be interesting to see Knockout's performance on more complex tasks like language modeling or multimodal learning, where missing modalities could be prevalent.

---

### Note · Authors · 2024-12-08

I have read and agree with the venue's withdrawal policy on behalf of myself and my co-authors.